# Early warning indicators of HIV drug resistance in the southern highlands region of Tanzania: Lessons from a cross-sectional surveillance study

**Samoel A. Khamadi**[1,2]\*, **Caroline Mavere**[1], **Emmanuel Bahemana**[1], **Anange Lwilla**[1], **Mucho Mizinduko**[1], **Seth Bwigane**[1], **Adela Peter**[1], **Joy Makando**[1], **Benjamin Peter**[1], **Patricia Agaba**[2,3], **Neha Shah**[2], **Boniphase Julu**[4], **Kavitha Ganesan**[2,3,5], **Peter Coakley**[2,4], **Elizabeth H. Lee**[2,6]

**1** HJF Medical Research International, Mbeya, Tanzania, **2** U.S. Military HIV Research Program, Walter Reed Army Institute of Research, Silver Spring, MD, United States of America, **3** Henry M. Jackson Foundation for the Advancement of Military Medicine, Bethesda, MD, United States of America, **4** Ifakara University, Ifakara, Tanzania, **5** Department of Epidemiology, Mailman School of Public Health, Columbia University, New York, NY, United States of America, **6** Uniformed Services University of the Health Sciences, Bethesda, MD, United States of America

\* skhamadi@kemri.go.ke

**Data Availability Statement:** All data are in the manuscript and/or supporting information files.

## Abstract

The World Health Organization early warning indicators (EWIs) permit surveillance of factors associated with the emergence of HIV drug resistance (HIVDR). We examined cross- and within-region performance on HIVDR EWIs for selected HIV care and treatment clinics (CTCs) in five regions of southern Tanzania. We retrospectively abstracted EWI data from 50 CTCs for the January to December 2013 period. EWIs included the following: on time ART pick-up, retention on ART, ARV stockouts, and pharmacy prescribing and dispensing practices. Data for pediatric and adult people living with HIV were abstracted from source files, and frequencies and proportions were calculated for each EWI overall, as well as stratified by region, facility, and age group. Across and within all regions, on average, on-time pick-up of pills (63.0%), retention on ART (76.0%), and pharmacy stockouts (69.0%) were consistently poor for the pediatric population. Similarly, on-time pill pick up (66.0%), retention on ART (72.0%) and pharmacy stockouts (53.0%) for adults were also poor. By contrast, performance on pharmacy prescribing and dispensing practices were as desired for both pediatric and adult populations with few facility-level exceptions. In this study, regions and facilities in the southern highlands of Tanzania reported widespread presence of HIVDR risk factors, including sub-optimal timeliness of pill pickup, retention on ART, and drug stockouts. There is an urgent need to implement the WHO EWIs monitoring to minimize the emergence of preventable HIV drug resistance and to maintain the effectiveness of first and second-line ART regimens. This is particularly critical in the context of new ART drug roll-out such as dolutegravir during the COVID-19 pandemic when resultant HIV service disruptions require careful monitoring, and for virologic suppression as countries move closer to epidemic control.

**Funding:** The study was funded by The US President's Emergency Plan for AIDS Relief (PEPFAR) under the cooperative agreement Cooperative Agreement # W81XWH-11-2-0174 to SAK. The funders had no role in study design, data collection and analysis, decision to publish, or preparation of the manuscript.

**Competing interests:** The authors have declared that no competing interests exist.

## Introduction

Despite many gains made in expanding access to lifesaving antiretroviral therapy (ART) and the resultant decline in HIV/AIDS related deaths in the first two decades of the twenty-first century, the absolute number of new HIV infections globally has continued to rise [1–3]. In 2020, there were more than 1.5 million new infections occurring worldwide and over 37.7 million people living with HIV (PLWH) [2].

Of concern, HIV drug resistance (HIVDR) is increasing and is compromising the effectiveness of ART across Africa [4]. HIV-1 drug resistance prevalence has been estimated for East Africa at 7.4% (95% CI: 4.3–12.7) eight years after ART roll-out between 2001–2011, with an annual increase estimated at 29% [5]. Pre-treatment HIVDR has risen to as high as 15% in countries such as Uganda [6]. In a cross-sectional study conducted in Kenya among HIV patients enrolled on ART between 2015–2017, HIVDR was reported to be as high as 82% in the sampled patients [7].

HIVDR mutations develop from ART pressure that results in viral rebound and treatment failure. ART adherence issues and suboptimal prescribing, dispensing, and counseling practices can lead to changes in the genetic structure of HIV that affect the ability of ART to block viral replication [8]. HIVDR in an individual can hamper the efficacy of ART regimens, and on a population-level, can pose a serious threat to national HIV program success. Improper use of ART and provision of suboptimal services at care and treatment clinics (CTC) can result in viral mutations that cause HIVDR [9]. Improper use of ART includes continuing certain regimens even when patients are failing treatment, and lack of timely adherence counseling for those who are non-adherent to their medication. Yet, HIVDR testing is not routinely performed as part of standard service delivery due to high costs associated with testing.

As an alternative to routine testing, the WHO early warning indicators (EWI) approach is a relatively inexpensive, non-laboratory method for large-scale program monitoring for the emergence of HIVDR [10]. EWIs are clinic, patient, and program factors that serve as a sentinel for HIVDR emergence [11]. WHO EWIs established in 2012 include: EWI-1 On-time pill pick-up: EWI-2 Retention on ART; EWI-3 pharmacy antiretroviral (ARV) stockouts; EWI-4 pharmacy prescribing and dispensing practices; and EWI-5: virologic suppression while on ART [12]. Early identification of these factors can help healthcare providers initiate appropriate corrective actions at the clinic-level, and may be used to alert policy-makers and program managers to take appropriate actions to contain and prevent HIVDR emergence at the national level [13].

In 2004, the United Republic of Tanzania implemented a policy of treating all eligible PLWH with ART, and by 2013, new infections and deaths decreased by more than 40% [14, 15]. To help protect these hard-won gains, the first and only national retrospective EWI study took place in 2010. This surveillance effort demonstrated that while physicians largely adhered to standards for prescribing ART, patient retention on ART was highly variable and lower than ideal, suggesting a brewing problem [16]. Unfortunately, no follow up EWI studies have been carried out nationally in Tanzania, leaving a critical knowledge gap.

Addressing EWIs as risk factors for HIVDR requires targeted understanding and intervention at facility-and regional-levels. Consequently, we studied selected WHO EWIs at health facilities in the southern highlands zone of Tanzania for the calendar year 2013 (S1 Table). This zone had the highest prevalence of HIV in Tanzania and a high total number of PLWH on treatment. We describe performance against WHO targets for the respective EWIs during 2013 and reflect on the continued relevance for combating HIV nearly ten years later.

## Methods

This cross-sectional, facility-based study was carried out in the regions of Mbeya, Songwe, Ruvuma, Rukwa and Katavi in the southern highlands of Tanzania. We retrospectively abstracted data for children (0–17 years) and adults (18 years and above) living with HIV who were enrolled in HIV care and treatment services at eligible health facilities. All facilities were rurally located and under the jurisdiction of the Tanzania Ministry of Health, Community Development, Gender, Elderly and Children (MoHCDEC) and receiving technical assistance through the U.S. Military HIV Research Program of Walter Reed Army Institute of Research, funded by the US President's Emergency Plan for AIDS Relief (PEPFAR). Facility inclusion criteria followed the WHO EWI protocol guidance: >3 years of experience with ART management; ≥30 newly enrolled patients on ART per fiscal quarter; and a range of health facility service delivery levels [17]. The latter included zonal and regional referral hospitals serving a catchment of at least 1,000,000 clients, district hospitals with at least 1 bed per 1,000 persons serving a catchment of 100,000–200,000, and health centers with a catchment of at least 50,000 persons. Client files were systematically sampled at each facility in accordance with WHO EWI sample size guidelines that are based on the number of clients enrolled on ART at a given facility over a 12-month period [18].

From July 2016 to August 2018, trained data clerks and CTC nurses abstracted data from January–December 2013 for EWIs 1–4 (EWI-1 On-time pill pick-up; EWI-2 Retention on ART; EWI-3 Pharmacy ARV stock out; EWI-4$^1$ Pharmacy Dispensing practices; and EWI-4$^2$ Pharmacy prescribing practices). The fifth indicator addressing virologic suppression after 12 months or more on ART was not evaluated as viral load monitoring was not routinely performed in Tanzania in 2013. Source documents consisted of ART registers, pharmacy stock records, and patient CTC medical records.

Table 1 provides definitions of the numerator and denominator for each indicator. Frequencies and proportions using all available data were calculated in Excel for each EWI overall, at the regional-level, and separately for each facility, and were then stratified for pediatric and adult populations. Denominators varied when calculating each EWI. WHO targets for each of the EWIs were used to grade performance against targets, and a stoplight color scheme was employed to visually convey desirable (green), fair (amber) or poor (red) performance. We did not compare EWI scores to one another as WHO targets differ for each EWI. Clinics with poor performance for any EWI were flagged for programmatic follow-up.

### Ethical approval

Research ethics approval was obtained from Mbeya Medical Research Ethics Committee (152/377), the National Health Research Ethics Committee at the National Institute for Medical Research Tanzania (NIMR 2083, 2015), and the Human Subjects Protections Branch of Walter Reed Army Institute of Research (WRAIR 2204, 2015). Consent was sought from each of the health facilities for use of the data in the study. The data that was used in the study was abstracted from patient files at the hospitals and anonymized before analysis. There was no contact made with the study participants. This applied to both the children and adult data.

## Results

### Facility characteristics

Fifty healthcare facilities with CTCs serving 15.6% (n = 18,668) of the total 119,429 patients on ART in the southern highlands met the inclusion criteria in 2013 (S1 Table). These included 33 health centers, 13 district hospitals, 3 regional referral hospitals and one zonal referral

**Table 1. The WHO early warning indicator definitions and targets as per 2012 EWI guidelines.**

| EWI | Definition | Numerator | Denominator | Target |
|-----|-----------|-----------|-------------|--------|
| EWI-1: On-time pill pick-up. | Proportion of patients (adult or children) that pick-up ART no more than two days late at the first pick-up after the baseline pick-up | Number of patients picking-up their ART "on time" at the first drug pick-up after baseline pick-up date. | Number of patients who picked-up ARV drugs on or after the designated EWI sample start date. | Desirable performance (green): >90%; fair performance (amber): 80–90%; poor performance (red): <80%. |
| EWI-2: Retention on ART. | Percentage of adults and children known to be alive and on ART 12 months after initiation. | Number of adults (or children) who are still alive and on ART 12 months after initiating treatment. | Total number of adults or children (excluding transfer outs) who initiated ART and were expected to achieve 12-month outcomes within the reporting period | Desirable performance (green): >85%; fair performance (amber); 75–80%; poor performance (red): <75% |
| EWI-3: Pharmacy stock-outs. | Percentage of months in a designated year in which there were no ARV drug stock-outs (both for adult and pediatric patients). | Number of months in the designated year in which there were no stock-out days of any ARV drug routinely used at the site. | 12 months of the reporting period. | Desirable performance (green): 100%; poor performance (red): <100%. |
| EWI-4[1]: Pharmacy dispensing practices | Percentage of patients (adults or children) being dispensed a mono- or dual-ART | Number of patients who pick-up from the pharmacy, a regimen consisting of one or two ARVs. | Number of patients picking up ART on or after the designated EWI sample start date | Desirable performance (green) defined as 0% patients picking-up a mono- or dual-ART; poor performance (red) defined as >0% patients picking-up a mono- or dual-ART. |
| EWI-4[2]: Pharmacy prescribing practices | Percentage of patients (adults or children) prescribing a mono- or dual-ART | Number of prescribed at the pharmacy with a regimen consisting of one or two ARVs. | Number of patients picking up ART on or after the designated EWI sample start date | Desirable performance (green) defined as 0% patients picking-up a mono- or dual-ART; poor performance (red) defined as >0% patients picking-up a mono- or dual-ART. |

hospital spread across urban and rural settings (Table 2). Sixteen (32%) facilities were located in the Mbeya region, 13 (26%) in Ruvuma region, 10 (20%) in Rukwa, seven (14%) in Songwe and the remaining 4 (8%) in Katavi region. Of these facilities, 34 (68%) provided primary care, 15 (30%) provided secondary care, and 1 (2.0%) provided tertiary care. Thirty-six (72%) of the

**Table 2. Summary of facility characteristics and records sampled by Southern Highlands Region.**

| | Region | | | | | |
|---|---|---|---|---|---|---|
| | Katavi (n = 4) | Mbeya (n = 16) | Rukwa (n = 10) | Ruvuma (n = 13) | Songwe (n = 7) | Total (n = 50) |
| **Variable** | n (%) | n (%) | n (%) | n (%) | n (%) | n (%) |
| **Total patients enrolled on ART in 2013** | 1285 (6.9) | 8850 (47.4) | 2213 (11.9) | 2859 (15.3) | 3461 (18.5) | 18668 (100.0) |
| **Total records reviewed** | | | | | | |
| Children (0–17 years) | 18 (3.6) | 201 (8.7) | 72 (6.4) | 123 (8.6) | 66 (7.2) | 480 (7.6) |
| Adults (> = 18 years) | 489 (96.4) | 2111(91.3) | 1052 (93.6) | 1315 (91.4) | 848 (92.8) | 5815 (92.4) |
| **Facility Level** | | | | | | |
| Primary[1] | 3 (75.0) | 10 (62.5) | 8 (80.0) | 10 (76.9) | 3 (42.9) | 34 (68.0) |
| Secondary[2] | 1 (25.0) | 5 (31.3) | 2 (20.0) | 3 (23.1) | 4 (57.1) | 15 (30.0) |
| Tertiary[3] | 0 (0.0) | 1 (6.3) | 0 (0.0) | 0 (0.0) | 0 (0.0) | 1 (2.0) |
| **Facility Ownership** | | | | | | |
| Government | 4 (100.0) | 11 (68.8) | 8 (80.0) | 8 (61.5) | 5 (71.4) | 36 (72.0) |
| Faith-based | 0 (0.0) | 5 (31.3) | 2 (20.0) | 5 (38.5) | 2 (28.6) | 14 (28.0) |
| Private | 0 (0.0) | 0 (0.0) | 0 (0.0) | 0 (0.0) | 0 (0.0) | 0 (0.0) |

[1] Primary: Health Centre; [2] Secondary: District or regional Referral Hospital; [3] Tertiary: Zonal Hospitals.

Note: Percentages are calculated and should be interpreted across all regions for the variable "Total patients enrolled on ART in 2013", and within each region for all other variables.

**Table 3. Overall EWI facility performance according to WHO targets (n = 50).**

| Indicator | Facility Performance n (%) | | |
|---|---|---|---|
| | Desirable | Fair | Poor |
| EWI-1: On-time pill pick-up (n = 50) | 0 (0.0) | 11 (22.0) | 39 (78.0) |
| EWI-2: Retention on ART (n = 50) | 5 (10.0) | 17 (34.0) | 28 (56.0) |
| EWI-3: Pharmacy ARV stock-out (n = 38) | 2 (5.3) | N/A | 36 (94.7) |
| EWI-4[1]: Pharmacy dispensing practices (n = 44) | 43 (97.7) | N/A | 1 (2.3) |
| EWI-4[2]: Pharmacy prescribing practices (n = 50) | 45 (90.0) | N/A | 5 (10.0) |

Abbreviations: EWI: Early Warning Indicator, n: count; %: percent; N/A: Not applicable. There were only two categories of performance for EWIs 3 and 4 (desirable and poor). For each EWI specific definition and performance range, please refer to Table 1 above.

facilities were government-owned under the jurisdiction of MoHCDEC, while the remaining were privately owned by faith-based organizations. Data for 6295 clients initiated on ART in 2013 were abstracted. Table 2 shows the details of the clients from whom data was abstracted [19].

## Overall EWI performance

Overall, the majority of facilities performed poorly on EWIs 1–3 using WHO performance targets (Table 3). None of the facilities met the WHO criteria for 'desirable' performance for EWI-1 (on-time pill pick-up). For EWI-2 (retention in care) and EWI-3 (pharmacy stockouts), only 10.0% of 50 facilities and 5.3% of 38 facilities with data had desirable performance. Conversely, for EWI-4[1] (dispensing practices), 97.7% out of 44 facilities with data had desirable performance while only a single facility had poor performance. Ninety percent of 50 total facilities had desirable performance on EWI-4[2] (prescribing practices).

## Regional EWI performance for children and adolescents

We stratified EWI performance by region (Table 4) and by individual facility (S2 Table) for pediatric ages 0–17 years. All five regions and the majority of facilities within the regions performed poorly on EWI-1 and EWI-3. For EWI-2, Songwe, Rukwa and Ruvuma had fair performance of 79.0%, 77.0% and 82.0% respectively, while Mbeya and Katavi performed poorly at 69.0% and 71.0%, respectively. All regions performed well on EWI-4[1] dispensing practices

**Table 4. Summary of EWI performance across all regions for pediatric ages 0–17 years.**

| Region | EWI-1 (On time pill pick-up) | EWI-2 (Retention on ART) | EWI-3 (ARV Pharmacy Stockout) | EWI-4[1] (Dispensing Practices) | EWI-4[2] (Prescribing Practices) |
|---|---|---|---|---|---|
| Mbeya | 59.0% | 69.0% | 69.0% | 0.0% | 0.0% |
| Songwe | 53.0% | 79.0% | 80.0% | 0.0% | 0.04% |
| Ruvuma | 72.0% | 82.0% | 67.0% | 0.0% | 0.0% |
| Rukwa | 75.0% | 77.0% | 78.0% | 0.0% | 0.0% |
| Katavi | 56.0% | 71.0% | 50.0% | 0.0% | 0.0% |
| All regions | 63.0% | 75.6% | 68.8% | 0.0% | 0.008% |

Note: 1. The colors reflect the WHO color codes for performance of each indicator in respective facilities (red-poor, amber- fair and green-desirable scoring). 2. Each score represents an average score of the performance of the facilities in each specific region.

and EWI-4[2] prescribing practices, except for Songwe that performed poorly on prescribing practices.

At the facility level, 10 out of 50 (20.0%) facilities had excellent performance (>90%) on EWI-1 (S2 Table). Of these 10 facilities, half were located in Rukwa. Nine (18.0%) clinics had fair performance (80–90%); and 31 (62%) performed poorly (<80%). For EWI-2, 12 (24%) facilities had excellent performance (>85%). Of these 12, five were located in Ruvuma. The remaining 20 (40.0%) and 18 (36.0%) facilities performed fairly or poorly, respectively. For EWI-3, eight of 38 (21.0%) facilities where data were successfully abstracted had desirable performance; the remaining performed poorly. For EWI-4[1] dispensing practices, all 44 (100.0%) facilities had desirable performance, while for prescribing practices, 49 out of 50 (98.0%) had desirable performance.

## Regional and facility-level EWI performance for adults ages 18 years and above

We also examined EWI performance by region (Table 5) and by individual facility (S3 Table) for adults ages 18 years and above. Performance was poor for EWI-1 and EWI-3 for all regions. For EWI-2, Ruvuma and Rukwa regions had fair performance at 76.0% and 75.0%, respectively, while all other regions performed poorly. Dispensing and prescribing practices were desirable for all regions.

No facility had desirable performance on EWI-1 (S3 Table). Twelve (24.0%) performed fairly, and the remaining (76.0%) performed poorly. For EWI-2, four (8.0%) had desirable performance, 18 (36.0%) had fair performance, and 28 (56.0%) had poor performance. For EWI-3, 4 out of 38 (10.5%) had desirable performance; the remaining clinics performed poorly. For EWI-4[1] dispensing practices, 43 out of 44 (97.7%) clinics had desirable performance, and one clinic in Mbeya had poor performance. For EWI-4[2] prescribing practices, 46 out of 50 (92.0%) ART clinics had desirable performance.

## EWI variability by age group

When stratified for adult versus pediatric age, EWIs 1–3 showed moderate variation across regions. For example, in each region, performance for on-time pill pick-up tended to be 5–10% higher for adults than children, except for Rukwa where this reversed. Performance on retention on ART was slightly lower for adults compared to children for all regions except Mbeya, although these differences may not be meaningfully different in practice. Interestingly, Songwe region had poor performance for retention on ART for adults, but fair performance

**Table 5. Summary of EWI performance across all regions for adults 18 years and above.**

| Region | EWI-1 (On time pill pick-up) | EWI-2 (Retention on ART) | EWI-3 (ARV Pharmacy Stockout) | EWI-4[1] (Dispensing Practices) | EWI-4[2] (Prescribing Practices) |
|---|---|---|---|---|---|
| Mbeya | 66.0% | 72.0% | 44.0% | 0.0% | 0.0% |
| Songwe | 65.0% | 70.0% | 29.0% | 0.0% | 0.0% |
| Ruvuma | 77.0% | 76.0% | 71.0% | 0.0% | 0.0% |
| Rukwa | 61.0% | 75.0% | 67.0% | 0.0% | 0.0% |
| Katavi | 62.0% | 68.0% | 56.0% | 0.0% | 0.0% |
| All regions | 66.2% | 72.2% | 53.4% | 0.0% | 0.0% |

Note: 1. The colors reflect the WHO color codes for performance of each indicator in respective facilities (red-poor, amber- fair and green-desirable scoring). 2. Each score represents an average score of the performance of the facilities in each specific region.

for children. Drug stockouts (EWI-3) were substantially worse for adult regimens than pediatric regimens in Mbeya, Songwe, and Rukwa regions in 2013. In Ruvuma and Katavi, pediatric stockouts were reported more frequently than for adult regimens, but not substantially so. There were no differences by age group for dispensing and prescribing practices.

## Discussion

Our study examined within- and across-regional performance on HIVDR EWIs for the 50 health facilities serving PLWH in the five regions of Mbeya, Ruvuma, Rukwa, Katavi and Songwe in the southern highlands of Tanzania. Across and within all regions, EWI-1 on-time pick-up of pills and EWI-3 pharmacy stock-outs were consistently poor overall and by adults and children. By contrast, EWI-4[1] performance on dispensing and EWI-4[2] prescribing practices was excellent overall by region and individual facilities. Ninety percent of facilities performed fair or poorly for EWI-2 retaining clients on ART, with little variation at the regional-level when stratified by age group.

### Across-and within-region performance

Poor performance for EWI-1 on-time pill pick up and EWI-3 pharmacy stockouts suggests systems-wide supply and demand challenges with ensuring medication is received on-time in the southern highlands for the study period, irrespective of facility management authority, level of services, or client age. On-time pill pick-up is an important measure of patient adherence and has previously been shown to be associated with poor patient-level outcomes including loss to follow up, development of HIVDR, virologic failure and death [19].

In the southern highlands, facilities serve large, rural catchment areas and clients often travel long distances to access care. Additionally, given the predominantly agrarian economy, many clients are occupied with farming activities for long stretches during planting season and may miss ART appointments. Policies that encourage multi-month scripting and dispensing can reduce the number of visits and thus missed appointments for pill pick-up [20]. They may also encourage retention on ART, which is intrinsically linked with on-time pill pick-up. Administratively, resource allocation decisions largely occur at the regional council-level in Tanzania, which could help explain the salient differences in performance of the different regions. In this study, it was noted that health facilities did not keep accurate records of inventories, resulting in delays to ordering drugs on time and pharmacy stockouts. This can be improved by training healthcare workers on better pharmacy inventory management, drug procurement procedures, and resource allocation for purchase of drugs [21].

Of all indicators, retention on ART showed the most variability at region and facility-level. Ruvuma and Rukwa regions, which included nearly half of participating facilities, had fair performance for retention on ART. We speculate that improved tracking and follow-up of patients in these regions where recordkeeping was anecdotally better at some of the facilities, in conjunction with follow-up by community health workers at home after missed appointments which has been shown to improve adherence, may have led to moderately better performance compared to other regions [22]. Provision of community-based adherence support and psychosocial support that includes home visits and adherence clubs has been shown to help improve adherence to ART for PLWH [22].

Across all regions, nearly all facilities had desirable ARV drug dispensing and prescribing practices for both adults and children, which reflects what has been previously reported in the literature for other African countries [23–26]. This nearly perfect performance indicates that clinical officers were providing PLWH with appropriate first-line medications. In the southern highlands, clinicians and pharmacists are routinely trained and mentored on ART drug

prescribing and dispensing, suggesting that program-level efforts to ensure clients receive the right drug regimens following national guidelines are effective. Although viral load testing is now standard of care in Tanzania, HIVDR testing for non-suppressed patients is not, thus reinforcing the importance of monitoring EWIs as population-level signals of HIVDR risk. Notably, dispensing practices are closely related with the emergence of HIVDR as dispensing of mono-or dual-ART and inappropriate dosing may lead to insufficient drug pressure that eventually results in the development of resistance to ART [27].

## Age-stratified performance variations

At the regional-level, performance results for adults were generally better for on-time pill pick-up and worse for retention on ART and drug stockouts as compared to pediatric ages. This is partially consistent with a Namibian EWI study that found a similar divide for retention on ART after 12 months, however that study found no difference between age groups for on-time pill pick-up and fewer stockouts for adult regimens compared to pediatric regimens [11]. While we cannot be certain of the reasons for age group differences, context, location and temporality matter. For example, stockouts are frequently driven by national shortages and may have a regional (geographic) component given the nature of supply chain distribution systems [28]. Of the five regions, Mbeya, Songwe, and Rukwa are grouped most centrally and adjacent to one another, so it is not surprising that they would be similar with respect to drug stockout patterns. Our results on EWIs 1–3 were consistent with rural performance in other studies from a similar time period, including one study of 10–19 year olds [27, 29]. Development of HIVDR at a young age can have deleterious implications as patients may run out of medication options to switch to later in life, making the performance on EWIs 1–3 overall and regionally for pediatric ages particularly alarming [30]. Contributing factors related to timely pick-up of medication and retention on ART for the pediatric population that have been reported elsewhere include delayed return of viral load results, inadequate adherence counseling skills and shortages of staff, all of which are modifiable [30].

## Targeting interventions to client, facility, and program

EWIs vary in terms of what they monitor, and resultantly, may require different approaches to address deficits. For example, EWIs 1 and 2 monitor client-side behavioral factors on drug pick-up and retention on ART, respectively, whereas EWIs 3 and 4 track facility and program-level indicators of ARV drug procurement and supply management, as well as client care through appropriate prescribing and dispensing practices [31]. Indicators may also be correlated, such as delayed ART pick-up and retention on ART after 12 months, which may necessitate synergistic interventions [16]. Our study facilities and catchment areas were spread out over five geographic regions where time, money and transportation may have all factored into clients' timeliness in seeking services. While client-side factors may not be totally insurmountable, evidence-based strategies put in place following this study to simultaneously address EWIs 1 and 2 included community healthcare worker outreach to clients who were not coming to the facilities to pick-up their drugs and multi-month dispensing of ART for eligible clients to reduce clinic visits [32–34].

Site-level factors contributing to pharmaceutical stockouts in Tanzania and elsewhere include poor inventory management practices, understaffing, competing activities, or inadequate training [25, 35]. Stockouts of ART can impact on-time pill pick-up and retention in care through delayed initiation of treatment, interrupted access to medication for clients, and disengagement of patients in care [36, 37]. In our study, we noted that participating health facilities did not keep consistent records of completed inventories which may have contributed

to delays in ordering drugs on time and subsequent pharmacy stockouts. Routinely training healthcare workers on proper pharmaceutical inventory management, drug procurement procedures, and resource allocation practices for purchase of drugs can all help to prevent stockouts [21]. Results of this study prompted an intentional effort in real-time to address stockouts in the southern highlands through training and mentorship of pharmacy staff and implementing standardized reordering systems with a minimum of three months' stock on hand to account for supply chain delays.

## Public health relevance

Monitoring EWIs at a program- or population-level continues to be a cost-effective and relevant method of HIVDR signal surveillance, and is a key facet of the WHO Global Strategy for HIV Drug Resistance Prevention and Assessment. WHO recommends yearly monitoring of EWIs of HIVDR at all ART clinics or a representative sample of all health facilities in a specific country. However, when this is not feasible in low resource settings, countries may alternatively consider national monitoring of EWIs every two or three years given the cost and time requirements. It is also important to put corrective measures in place and monitor their implementation before doing another EWI. In the absence of a national strategy, countries can adopt facility-based monitoring. Health facilities can include this activity in their quality assurance procedures and conduct EWI as part of their performance improvement plans.

Additionally, the 2021 iteration of the WHO Global Strategy encourages countries like Tanzania to adapt, adopt and implement a national version in line with its key tenets: monitoring EWIs annually, and implementing HIVDR surveys. While Tanzania has yet to do so, the opportunity and rationale persists to integrate annual EWI monitoring into a national plan. Paired with our regional findings, the 2010 national study could provide a framework and basis for operationalizing EWI monitoring as a routine, annual activity on a national scale. Tanzania has a robust electronic reporting system at health facility level which includes HIV service delivery indicators. Integration of the set of EWIs plus standardized annual reporting could be an effective system for following this crucial safety signal for HIVDR. Adopting a similar approach to reporting through the national DHIS2 or other health information system may be a practical consideration for other countries in the absence of a unified national plan.

Further, the roll-out of new first-line drugs such as dolutegravir beginning in 2020 is promising, but requires careful monitoring to protect long-term effectiveness at a population-level [23]. EWIs are a readily available solution. While we saw reliable performance on prescribing and dispensing practices in 2013, new ART regimens will necessitate updates to policy and guidelines over time. Ongoing monitoring of EWI signals should coincide with shifts in Tanzanian guidelines and regimen availability in-country, in line with globally accepted standards. At an individual-level, we are aware of a single cross-sectional pediatric study documenting client-level HIVDR that coincided with roll-out of pediatric dolutegravir 10 mg (pDTG) in the southern highlands of Tanzania, alongside a sister study in Kenya [38]. As cost of routinizing HIVDR testing is prohibitive outside of research, program- and population-level monitoring for HIVDR signals remains a cost-effective, feasible alternative.

The COVID-19 pandemic has placed tremendous strain globally on healthcare service delivery and has led to disruptions in routine HIV health services, with the potential for downstream outcomes including HIV viral non-suppression and development of resistance [39, 40]. During pandemics, monitoring of EWI-1 and EWI-2 in particular continues to be a viable tool for tracking population-level disruption of services, and provides strategic information for policy and program decision-making. Although much of Africa avoided the level of morbidity and mortality from COVID-19 that Europe and North America faced early on, robust HIVDR

surveillance systems that include EWIs or similar cost-effective signals can help inform and mitigate the effects of future pandemics on the HIV response.

As Tanzania moves toward epidemic control, maintaining an undetectable viral load in the individual has become a cornerstone of HIV programming. While previously donors such as PEPFAR emphasized finding and putting all PLWH on ART, emphasis on ensuring all PLWH are virally suppressed has recently become a renewed, strategic focus of program planning and funding [41]. EWI performance monitoring can serve as a bellwether to identify geographic regions, localities or facilities needing intervention to improve care quality and reduce risk factors for HIVDR. Health facilities can develop simple tools for tracking these indicators at facility level to monitor performance and design appropriate, targeted interventions to improve quality of services in the ART clinic setting. Healthcare workers should be trained to abstract and analyze data routinely to serve as a means of assuring quality of care. Further, monitoring EWIs such as prescribing practices and on-time pill pickup can help ensure that an aging PLWH population facing comorbidities is receiving standard of care. This focus on quality should lead to improved health outcomes, and reciprocally, reduce risk for viral non-suppression in the individual [42]. This study aligns well with the WHO Global Strategy to prevent and minimize the emergence of HIV drug resistance as it provides a simple yet effective way to monitor for emergence of HIV drug resistance through promotion of best practices and identifying suboptimal or ineffective practices at health facilities that can result in development of HIV drug resistance.

## Limitations

Our study had several limitations. Data used in this study are from January to December 2013. Per the protocol approved in 2016, we followed the 2013 sampling frame for abstraction of complete data sets for patients at the CTCs that qualified for the EWI study were those who had visited the facilities consistently for at least 12 complete months. This lag time limits the ability to extrapolate and generalize conclusions of the findings to the Southern Highlands today. Nonetheless, the variation in performance we saw highlights how important tracking EWIs on a granular level may be and indicates how urgent tracking EWIs as a safety signal may be. Additionally, the findings can provide a benchmark for comparison with future EWI monitoring or HIVDR surveys, such as findings from the aforementioned cross-sectional pediatric HIVDR study from the Southern Highlands, in an otherwise limited landscape of evidence. Notably, historical practices of the past years continue to greatly impact development of HIV drug resistance today; a reminder that downstream prevention of HIV drug resistance in future requires foresight and planning now.

Some missing data for EWIs 3 (stockouts) and 4[1] (dispensing practices) may have introduced bias to regional results for these indicators. For example, of 12 facilities with missing data for EWI-3, five were located in the Mbeya region, resulting in 31.3% of Mbeya facilities that did not have records of whether there were pharmacy stockouts. Likewise, for EWI-4, of 6 facilities with missing data, five were also located in the Mbeya region. Missing data included incomplete documentation of ART drug regimen in source documents, the number of days dispensed, and the next date of pickup of drugs. While we cannot be certain, missing data may be related to electronic record system unavailability in 2013, understaffing, or training gaps. It is plausible that missing or incomplete data at these facilities could suggest these facilities might have had poor performance for these EWIs. Nonetheless, identification of documentation challenges permits targeted intervention with facility staff as a first step.

Notably, the EWIs were updated to include several additional indicators as of 2016: loss to follow up at 12 months, on-time appointment keeping, and viral load completion. Further,

viral load monitoring was not routine in Tanzania in 2013 and thus could not be reliably assessed. However, definitions and procedures for reporting EWIs 1–4 have not changed significantly, and thus our findings for 2013 continue to be the best available reflection of EWI-related program quality for that time period and thereafter in terms of indicating what may be driving HIVDR patterns in the southern highlands' region. Future efforts to monitor regional EWIs should include the additional EWI program quality indicators, in line with the current WHO Global Strategy.

Although we followed the WHO protocol for facility inclusion, this study was not a census of all facilities and therefore regional results may not be generalizable to all facilities. The 'true' within and across regional performance may more closely reflect the performance of larger facilities included in our sample which is useful for program planning purposes from a volume and resource allocation standpoint. However, regional results cannot reliably be applied to smaller facilities that did not meet inclusion criteria, as there may be unknown facility-level factors that could affect performance.

## Conclusion

In our retrospective study of Tanzanian data from 2013, there was widespread suboptimal performance for EWIs used to monitor on-time ART pick-up, retention on ART, and drug stock-outs for southern highlands regions and health facilities. Stratifying by age indicated some variation, although results generally remained poor overall. Promisingly, regionally aggregated and facility-level performance on pharmacy prescribing and dispensing practices was desirable. Poor programmatic performance on EWIs can facilitate the emergence of HIVDR at a population-level. As new pharmaceutical interventions such as dolutegravir are rolled-out and challenges for continuity of care like the COVID-19 pandemic are faced, continuous monitoring of risk factors for development of HIVDR remains critical.

## Supporting information

**S1 Table. Details of facilities from which clients were enrolled.**
(DOCX)

**S2 Table. Individual facility performance for pediatric population.**
(DOCX)

**S3 Table. Individual facility performance for the adult population.**
(DOCX)

## Acknowledgments

The study acknowledges the support of health facility staff from all the hospitals who supported the study team with data collection and cleaning.

**Disclaimer:** The views expressed are those of the authors and should not be construed to represent the positions of the U.S. Army, the Uniformed Services University of the Health Sciences, HJF or the Department of Defense.

## Author Contributions

**Conceptualization:** Samoel A. Khamadi, Anange Lwilla, Mucho Mizinduko, Seth Bwigane, Adela Peter, Kavitha Ganesan.

**Data curation:** Samoel A. Khamadi, Caroline Mavere, Emmanuel Bahemana, Anange Lwilla, Seth Bwigane, Benjamin Peter, Patricia Agaba, Boniphase Julu, Elizabeth H. Lee.

**Formal analysis:** Samoel A. Khamadi, Caroline Mavere, Emmanuel Bahemana, Anange Lwilla, Mucho Mizinduko, Seth Bwigane, Adela Peter, Joy Makando, Neha Shah, Boniphase Julu, Kavitha Ganesan, Peter Coakley, Elizabeth H. Lee.

**Funding acquisition:** Samoel A. Khamadi.

**Investigation:** Samoel A. Khamadi, Caroline Mavere, Emmanuel Bahemana, Mucho Mizinduko, Seth Bwigane, Kavitha Ganesan.

**Methodology:** Samoel A. Khamadi, Caroline Mavere, Emmanuel Bahemana, Anange Lwilla, Mucho Mizinduko, Joy Makando, Elizabeth H. Lee.

**Project administration:** Samoel A. Khamadi, Caroline Mavere, Seth Bwigane, Adela Peter.

**Resources:** Samoel A. Khamadi, Caroline Mavere.

**Software:** Samoel A. Khamadi.

**Supervision:** Samoel A. Khamadi, Caroline Mavere, Emmanuel Bahemana, Mucho Mizinduko, Seth Bwigane.

**Validation:** Samoel A. Khamadi, Caroline Mavere, Emmanuel Bahemana, Anange Lwilla, Seth Bwigane, Benjamin Peter, Neha Shah, Elizabeth H. Lee.

**Visualization:** Samoel A. Khamadi, Seth Bwigane.

**Writing – original draft:** Samoel A. Khamadi, Caroline Mavere, Emmanuel Bahemana, Anange Lwilla, Mucho Mizinduko, Seth Bwigane, Adela Peter, Joy Makando, Benjamin Peter, Patricia Agaba, Neha Shah, Boniphase Julu, Kavitha Ganesan, Peter Coakley, Elizabeth H. Lee.

**Writing – review & editing:** Samoel A. Khamadi, Caroline Mavere, Emmanuel Bahemana, Anange Lwilla, Mucho Mizinduko, Seth Bwigane, Adela Peter, Joy Makando, Benjamin Peter, Patricia Agaba, Neha Shah, Boniphase Julu, Kavitha Ganesan, Peter Coakley, Elizabeth H. Lee.

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
