## [Decision Letter · Decision Letter 0]

10 Nov 2022

PGPH-D-22-01203

EARLY WARNING INDICATORS AT FACILITIES IN THE SOUTHERN HIGHLANDS REGION OF TANZANIA: LESSONS FOR HIV DRUG RESISTANCE SURVEILLANCE

Dear Dr. Khamadi,

Thank you for submitting your manuscript to PLOS Global Public Health. After careful consideration, we feel that it has merit but does not fully meet PLOS Global Public Health’s publication criteria as it currently stands. Therefore, we invite you to submit a revised version of the manuscript that addresses the points raised during the review process.

We look forward to receiving your revised manuscript.

Kind regards,

Patrick A. Palmieri, DHSc, EdS, MBA, MSN, PGDip(Oxon), ACNP, RN, CPHRM, CPHQ, FFNMRCSI, FAAN

Academic Editor

Journal Requirements:

1. There are 15 authors listed on this article but the author contributions are note listed at the close of the manuscript with the other disclosures. When returning the manuscript, please provide the information specific to the contributions of each author per the PLOS requirements (https://journals.plos.org/plosone/s/authorship#loc-author-contributions).

2. Please send a completed 'Competing Interests' statement, including any COIs declared by your co-authors. If you have no competing interests to declare, please state "The authors have declared that no competing interests exist". Otherwise please declare all competing interests beginning with the statement "I have read the journal's policy and the authors of this manuscript have the following competing interests:"

3. Please amend your detailed Financial Disclosure statement. This is published with the article. It must therefore be completed in full sentences and contain the exact wording you wish to be published.

4. We notice that your supplementary tables are included in the manuscript file. Please remove them and upload them with the file type 'Supporting Information'. Please ensure that each Supporting Information file has a legend listed in the manuscript after the references list.

Editor Comments:

Thank you for submitting your work to PLOS Global Public Health. We waited for a reviewer to send a report which slowed the process for making an editorial decision.

Please provide the completed STROBE checklist for cross-sectional studies (https://www.equator-network.org/reporting-guidelines/strobe/) with the page and line numbers for each reporting element noted in the checklist. Please make sure to review the supporting document with the criteria and rationale for each element. There are multiple elements not reported or partially reported in the current manuscript.

In addition, the dates for the data collection of the data used for the current study are not clearly presented. In the abstract, the data underlying this study was stated as "In 2016, we retrospectively collected EWI data from 50 CTCs for the January to December 2013 period." Then, the manuscript states, "From July 2016 to August 2018, trained data clerks and CTC nurses abstracted data from January – December 2013 for EWIs 1-4 (EWI-1 On-time pill pick-up; EWI-2 Retention on ART; EWI-3 Pharmacy ARV stock out; EWI-4 Pharmacy Dispensing practices; and EWI-4 Pharmacy prescribing practices)." The data for this study was not collected from 2016 to 2018, instead the data was abstracted. As such, the data is nearly ten year old. For this reason, the relevancy of the data needs to be questioned in 2022. Please explain how this data is relevant. When responding to the editor and reviewer feedback, please provide detailed information about other studies, if any, that were published with this data.

Reviewers' comments:

Reviewer's Responses to Questions

**Comments to the Author**

1. Does this manuscript meet PLOS Global Public Health’s publication criteria? Is the manuscript technically sound, and do the data support the conclusions? The manuscript must describe methodologically and ethically rigorous research with conclusions that are appropriately drawn based on the data presented.

Reviewer #1: Yes

Reviewer #2: Yes

2. Has the statistical analysis been performed appropriately and rigorously?

Reviewer #1: Yes

Reviewer #2: Yes

3. Have the authors made all data underlying the findings in their manuscript fully available (please refer to the Data Availability Statement at the start of the manuscript PDF file)?

Reviewer #1: Yes

Reviewer #2: Yes

4. Is the manuscript presented in an intelligible fashion and written in standard English?

Reviewer #1: Yes

Reviewer #2: Yes

5. Review Comments to the Author

Reviewer #1: The research looks into an important issue of public health concern, as well as a vital aspect of HIV - HIV drug resistance. Using EWIs as risk factors is a convenient, cost-effective and simple tool proposed by WHO. Methods are clearly explained. Implementation is according to methods. Data sets are available with results. Limitation to full application of all EWIs are clearly explained. Well discussed and concluded in simple comprehensible English.

Reviewer #2: Thank you for this timely and important study on the role of HIVDR EWI surveillance systems. Please respond to the following comments:

1. P. 6 Line 42. In this manuscript you provide an important rationale for routine EWI as it relates HIVDR and your research confirms this. However, in the abstract the language does not convey the urgency of implementing WHO EWIs.

Suggested change:

There is an urgent need to implement WHO EWI to minimize the emergence of preventable HIV drug resistance and to maintain the effectiveness of first- and second-line ART regimens. This is particularly critical in the context of new ART drug roll-out such as dolutegravir, during the COVID-19 pandemic when resultant HIV service disruptions require careful monitoring, and for virologic suppression as countries move closer to epidemic control.

2. In discussion, please indicate how this study aligns with WHO Global Strategy to further strengthen the significance of this study.

This is taken from the strategy statement: “WHO has developed a Global Strategy for HIV Drug Resistance Prevention and Assessment using a public health approach (3). This strategy supports optimal functioning of treatment programmes to minimize emergence of preventable HIV drug resistance and to maintain the effectiveness of first- and second-line ART regimens. WHO recommends that countries adapt and implement a national strategy, based on the Global Strategy, to assess and minimize emergence of preventable HIV drug resistance.”

3. On P7.Line 82, you state: ‘To help protect these hard-won gains, the first national retrospective EWI study took place in 2010’. Can you please include more information about whether this has continued. If not then this is an opportunity to directly call on the TZ government and others to recommit efforts to routine EWI.

4. Can you tell us in the text why you selected January – December 2013 as opposed to a more recent period? Is the age of the data a limitation that needs to be addressed in the limitations section?

4. In the intro or discussion, can you provide more details about frequency of conducting EWI. What does WHO recommend? Do you agree? In the absence of a national strategy what would you suggest for others outside of TZ?

6. PLOS authors have the option to publish the peer review history of their article (what does this mean?). If published, this will include your full peer review and any attached files.

**Do you want your identity to be public for this peer review?** For information about this choice, including consent withdrawal, please see our Privacy Policy.

Reviewer #1: **Yes: **Mekolle Enongene Julius

Reviewer #2: No

---

## [Decision Letter · Decision Letter 1]

28 Feb 2023

EARLY WARNING INDICATORS OF HIV DRUG RESISTANCE IN THE SOUTHERN HIGHLANDS REGION OF TANZANIA: LESSONS FROM A CROSS-SECTIONAL SURVEILLANCE  STUDY

PGPH-D-22-01203R1

Dear Dr. Khamadi,

We are pleased to inform you that your manuscript 'EARLY WARNING INDICATORS OF HIV DRUG RESISTANCE IN THE SOUTHERN HIGHLANDS REGION OF TANZANIA: LESSONS FROM A CROSS-SECTIONAL SURVEILLANCE  STUDY' has been provisionally accepted for publication in PLOS Global Public Health.

Best regards,

Julia Robinson

Executive Editor

Reviewer Comments (if any, and for reference):

Reviewer's Responses to Questions

**Comments to the Author**

1. If the authors have adequately addressed your comments raised in a previous round of review and you feel that this manuscript is now acceptable for publication, you may indicate that here to bypass the “Comments to the Author” section, enter your conflict of interest statement in the “Confidential to Editor” section, and submit your "Accept" recommendation.

Reviewer #2: All comments have been addressed

2. Does this manuscript meet PLOS Global Public Health’s publication criteria? Is the manuscript technically sound, and do the data support the conclusions? The manuscript must describe methodologically and ethically rigorous research with conclusions that are appropriately drawn based on the data presented.

Reviewer #2: Yes

3. Has the statistical analysis been performed appropriately and rigorously?

Reviewer #2: Yes

4. Have the authors made all data underlying the findings in their manuscript fully available (please refer to the Data Availability Statement at the start of the manuscript PDF file)?

Reviewer #2: Yes

5. Is the manuscript presented in an intelligible fashion and written in standard English?

Reviewer #2: Yes

6. Review Comments to the Author

Reviewer #2: The authors have addressed my prior comments/questions.

7. PLOS authors have the option to publish the peer review history of their article (what does this mean?). If published, this will include your full peer review and any attached files.

**Do you want your identity to be public for this peer review?** For information about this choice, including consent withdrawal, please see our Privacy Policy.

Reviewer #2: No
